# Association of pharmacotherapy with all-cause mortality among patients with irritable bowel syndrome
Sepideh Mehravar[1], Yee Hui Yeo [1], Mark Pimentel[1], Parnian Naji[2], Wee Han Ng[3], Nils Burger[4,5], Will Takakura[6] & Ali Rezaie [1] ✉

## Abstract

**Background** Irritable bowel syndrome (IBS) is a common disorder associated with high healthcare costs and reduced quality of life. The long-term safety of its pharmacotherapies remains unclear. This study aims to assess the association between long-term pharmacotherapies and all-cause mortality in this population.

**Method** We performed a retrospective cohort study using a nationwide U.S. electronic health record database (January 1, 2005, to January 1, 2023). A 1:1 propensity score-matched cohort included 669,083 adults (aged 18–65) with IBS. Patients were grouped by pharmacotherapy use, with subgroup analyses for IBS with diarrhea (IBS-D) and IBS with constipation (IBS-C). Follow-up started at the time of medication prescription after diagnosis. Exposures included guideline-recommended therapies for IBS, IBS-D, or IBS-C. The primary outcome was all-cause mortality, assessed using Cox proportional hazards models and target trial emulation.

**Results** Antidepressant use is associated with an increased risk of all-cause mortality (hazard ratio [HR], 1.35; 95% CI, 1.26–1.45; mortality rate, 1.6% vs. 1.0%). This association remains consistent across antidepressant subclasses and demographic subgroups. Antispasmodic use is not linked to increased mortality (HR, 0.95; 95% CI, 0.89–1.00). For IBS-D, cholestyramine/colestipol, eluxadoline, and rifaximin are not associated with mortality. However, diphenoxylate (HR, 1.89; 95% CI, 1.02–3.51) and loperamide (HR, 2.39; 95% CI, 1.48–3.90) show increased mortality risk. For IBS-C, polyethylene glycol-3350 and secretagogues have no significant association with mortality.

**Conclusions** These findings raise concerns regarding the safety of antidepressants and mu receptor agonists in IBS treatment and underscore the need for cautious prescribing and further research.

## Plain language summary

Irritable bowel syndrome (IBS) is a common digestive disorder that can greatly affect quality of life. Many patients use long-term medications to manage symptoms, but the safety of these treatments over time is not well understood. We used a large U.S. electronic health record database to study adults with IBS and compared death rates among those taking different IBS medications. We found that antidepressants and certain antidiarrheal drugs, such as loperamide and diphenoxylate, were linked to a higher risk of death. Other medications, including antispasmodics and treatments for constipation, were not associated with this risk. These findings highlight the importance of careful prescribing and further investigation into the long-term safety of IBS medications.

Irritable bowel syndrome (IBS) is a chronic functional gastrointestinal disorder characterized by abdominal pain associated with altered bowel habits, which can present as diarrhea-predominant (IBS-D), constipation-predominant (IBS-C), or a mixed pattern (IBS-M). The pathophysiology of irritable bowel syndrome (IBS) is heterogeneous, involving a complex interplay of altered gut-brain signaling, visceral hypersensitivity, dysbiosis, immune activation, and psychosocial factors, which vary widely among individuals[1,2].

The global burden of IBS is substantial, with prevalence rates estimated at 10–15% of the population worldwide[3]. In the United States, similar figures are observed, translating to significant healthcare resource utilization and economic costs, as well as reduced quality of life for patients[4]. IBS

[1]Karsh Division of Gastroenterology and Hepatology, Department of Medicine, Cedars-Sinai, Los Angeles, CA, USA. [2]Department of Surgery, University Hospitals Cleveland Medical Center, Cleveland, OH, USA. [3]Department of Public Health and Primary Care, University of Cambridge, Cambridge, UK. [4]Department of Cancer Biology, Dana-Farber Cancer Institute, Boston, MA, USA. [5]Department of Cell Biology, Harvard Medical School, Boston, MA, USA. [6]Department of Medicine, Division of Gastroenterology and Hepatology, Kaiser Permanente Panorama City, Panorama City, CA, USA. ✉e-mail: ali.rezaie@cshs.org

disproportionately affects younger adults and women, further contributing to its societal impact.

Several medications have been approved by the U.S. Food and Drug Administration (FDA) for the treatment of IBS. These include lubiprostone, tenapanor, linaclotide, and plecanatide for IBS-C[5], and rifaximin, eluxadoline, and alosetron for IBS-D[6–8]. However, fewer than 20% of patients with IBS-C and fewer than 10% of patients with IBS-D report being treated with FDA-approved medications[9]. In addition to FDA-approved options, societal guidelines and publications advocate for the use of other medication classes, including tricyclic antidepressants (TCAs), selective serotonin reuptake inhibitors (SSRIs), serotonin-norepinephrine reuptake inhibitors (SNRIs), antispasmodics, and bile acid sequestrants[10–14]. These agents, while often prescribed off-label, address various symptom domains of IBS.

In this nationwide retrospective cohort study, long-term use of certain IBS pharmacotherapies is associated with increased all-cause mortality risk. Antidepressant use in IBS is associated with a higher risk of all-cause mortality across subclasses and patient subgroups, whereas antispasmodics are not associated with increased mortality. Among patients with IBS-D, μ-opioid receptor agonists, including loperamide and diphenoxylate, are associated with increased mortality risk, while rifaximin and bile acid sequestrants are not. For IBS-C, commonly used laxatives and secretagogues show no significant association with mortality. These findings highlight important safety differences among commonly prescribed IBS medications and underscore the need for cautious long-term prescribing and further prospective investigations.

## Methods
### Data acquisition
The study data was collected and analyzed on December 15, 2024, using TriNetX Analytics, which compiles de-identified EHR data from 106 U.S. healthcare organizations[15], primarily large academic medical centers, spanning all 50 states. It includes medical and pharmaceutical claims from commercial and governmental payers, covering diverse demographics. Our analysis utilized aggregated EHRs from 142.9 million patients.

TriNetX categorizes race and ethnicity data from the participating healthcare systems into the following classifications: (1) Race: Asian, American Indian or Alaska Native, Black or African American, Native Hawaiian or Other Pacific Islander, White, unknown; and (2) Ethnicity: Hispanic or Latino, non-Hispanic or Latino, unknown ethnicity.

### Ethical considerations
TriNetX complies with HIPAA and its integrated tools, including incidence, prevalence, outcomes, survival analysis, and PSM, enable patient-level analyses while presenting only population-level data.

This study was conducted in accordance with the Declaration of Helsinki. The TriNetX Research Network aggregates de-identified electronic health record data from participating healthcare organizations. Each contributing site is responsible for obtaining institutional review board (IRB) approval, informed consent, and permission for secondary use of data, as required by local regulations.

The current study involved secondary analysis of de-identified data and was therefore determined to be exempt from IRB review and informed consent requirements, in accordance with institutional policies and U.S. federal regulations for research using de-identified data.

### Study population
We included the entire group of patients with IBS, IBS (International Classification of Diseases, 10th Revision (ICD-10 code) K58), IBS with diarrhea (IBS-D, ICD-10 code K58.0), IBS with constipation (IBS-C, ICD-10 code K58.1), unspecified IBS (ICD-10 code K58.9), mixed IBS (ICD-10 code K58.2), and other forms of IBS (ICD-10 code K58.8)[16]. The study protocol, encompassing inclusion and exclusion criteria, target trial emulation, eligibility criteria, exposure definition, secondary outcome definition,

negative control definition, and covariate definitions, is detailed in Supplementary Table 1 and Supplementary Data 1 and 2. All cohorts had no recorded death before the index event.

### Determination of medication use pattern
The *Treatment Pathways* feature in the TriNetX database which allows for the identification of top treatments within a cohort was utilized to determine the prevalence and the order of use of recommended medications in IBS. This approach enabled us to determine the real-world usage of medication classes in IBS to initiate the analysis for all-cause mortality between 2005 and 2023 as described below.

### Medication classes used in all IBS subtypes
Antispasmodics. We assessed mortality rates in patients prescribed dicyclomine or hyoscyamine. The index event was the diagnosis of IBS with an antispasmodic prescription. The control group consisted of IBS patients who were never prescribed antispasmodics.

Antidepressants. To evaluate the mortality rates among patients prescribed antidepressants, including SSRIs (i.e., citalopram, sertraline, escitalopram, fluoxetine, fluvoxamine, paroxetine), TCAs (i.e., amitriptyline, imipramine, desipramine, nortriptyline, doxepin, trimipramine), SNRIs (i.e., venlafaxine, duloxetine, milnacipran), and mirtazapine, with at least two refills[17] of each medication (Fig. 1,) later expanded to 1, 4, 8, 12, 16, and 20 refills to evaluate the impact of cumulative exposure on mortality patterns. These specific antidepressants were chosen for this study because they are recommended as second-line treatment options for managing IBS, as outlined in various societal guidelines and resources[10,11,13,14]. The index event was the diagnosis of IBS with an antidepressant prescription. The control group consisted of IBS patients who were never prescribed antidepressants.

A subgroup analysis was conducted for SSRIs, SNRIs, TCAs, and mirtazapine, with each group compared to patients with IBS who were never prescribed antidepressants (non-users) as controls.

### Medication classes used in IBS-D subtype
For IBS-D, the study included patients who were prescribed antidepressants, mu receptor agonists, antispasmodics, bile acid sequestrants, eluxadoline, or rifaximin, following clinical guidelines[10,12,13]. Importantly, participants in this group had no history of antidepressant prescriptions to minimize selection bias. The index event was IBS-D diagnosis and medication initiation. The control group included patients with IBS-D who were never prescribed antidepressants or IBS-D medications. Each medication subgroup was analyzed by comparing treated patients to those never prescribed the respective medication or antidepressants.

### Medication classes used in IBS-C subtype
For IBS-C, we included patients who were prescribed antidepressants, polyethylene glycol 3350 (PEG-3350), antispasmodics (hyoscyamine and dicyclomine), and secretagogues (tenapanor, plecanatide, lubiprostone, and linaclotide) per clinical guidelines[10,11,13]. Participants in this group had no history of antidepressant use. The index event was IBS-C diagnosis and medication initiation. The control group included patients with IBS-C who were never prescribed antidepressants or IBS-C medications. Each medication subgroup was analyzed by comparing treated patients to those never prescribed the respective medication or antidepressants.

Primary outcome. All-cause mortality was assessed between 6 months and 15 years after the index event, comparing PSM users and non-users based on ICD-10 code R99 ('mortality') in EHRs[18]. The follow-up period ranged from 6 months to 10 years, considering FDA approval date variability for IBS-D and IBS-C medications.

Secondary outcome. To investigate potential causes of mortality, we performed a subgroup analysis by excluding participants with specific

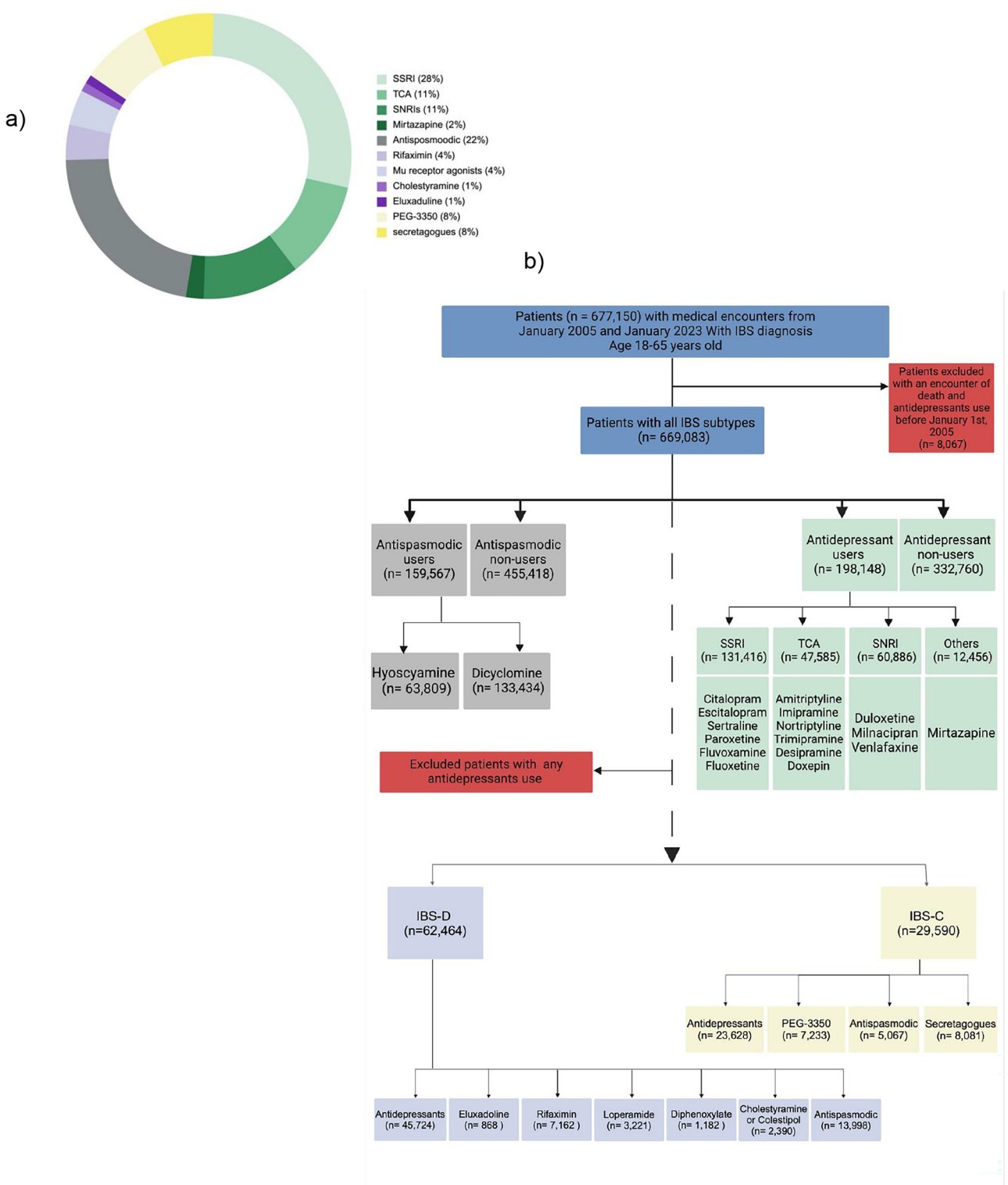

**Fig. 1 | Study flow diagram. a** Medication use pattern among all patients with IBS. Antidepressants followed by antispasmodics are the most used medications in patients with IBS. **b** Study flow diagram for assessment of all-cause mortality in patients with IBS and in patients with IBS-D and IBS-C. Patients receiving antidepressants were excluded from IBS-D and IBS-C cohorts given the association of antidepressant users with increased all-cause mortality in all IBS patients in the preceding analysis.

diagnoses (supplementary Table b) prior to the index event and following them from 6 months up to 15 years. This comprehensive approach enabled us to evaluate specific outcomes while controlling for preexisting conditions that could confound the results.

**Specification of target trials and active comparator.** A target trial emulation framework was utilized to evaluate the comparative efficacy of antidepressants versus antispasmodics in patients with IBS. As antispasmodics are recommended as therapeutic alternatives in multiple

clinical guidelines[10,11] and considered a first-line treatment, this cohort was deemed an appropriate active comparator. The target trial simulation was extended to include subgroup analyses, where exposure to SSRIs, TCAs, SNRIs, and mirtazapine was compared to antispasmodic[19,20]. Similarly, exposure to loperamide or diphenoxylate was compared to antispasmodics in IBS-D.

**Negative control outcomes.** Negative controls are characterized as results that deviate from the tested outcome and are highly unlikely to be directly associated with the intervention in question. Therefore, if the association is absent, it can be concluded that the relationship between the variable and the outcome of interest is also unbiased[21] (Supplementary Table c).

**Analysis approach.** For each study population, medication users and non-users were 1:1 PSM using nearest-neighbor greedy matching (caliper = 0.1 SMD)[22] on 56 covariates that are potential risk factors that could affect IBS or mortality[23] (Supplementary Data 2). The causal estimates of interest signify the intention-to-treat effect of assignment to the treatment. Cumulative occurrences were calculated utilizing the Kaplan–Meier survival analysis, and Cox proportional hazards analyses were employed to compare daily rates of time-to-event during the follow-up period following the index incident. Hazard ratios (HRs) and 95% confidence intervals (CIs) were computed to describe the relative hazard of the outcomes based on a comparison of time-to-event rates. Continuous variables are presented as mean ± standard deviation (SD). Subgroup analyses were performed in patients stratified according to sex, age subgroups (18–40, 40–65), racial and ethnic classifications, and BMI ($18 < BMI < 25$ and $BMI \geq 25$ units).

Data visualization and statistical analysis were conducted using Prism 10.0 (GraphPad, USA). A $p$-value < 0.05 was considered significant, with significance levels denoted as $*p < 0.05$. Figures. were created using Adobe Illustrator 2022, GraphPad Prism 10.0, R (v4.2), and BioRender.

# Results
## Medication classes used in IBS and its subtypes
The study included 669,083 patients with IBS who met the inclusion criteria. Antidepressants (52.3%), followed by antispasmodics (22.1%), were the most commonly used medications (Fig. 1a). Cohort creation processes to assess all-cause mortality in patients with IBS, IBS-D, and IBS-C are illustrated in Fig. 1b. The follow-up duration for each cohort is detailed in Supplementary Table 2.

## All-cause mortality associated with medication classes used in all IBS subtypes
**All-cause mortality associated with antidepressant use.** Among antidepressant users, 78% were female with a mean age of 38.13 ± 12.6 (SD), compared to 74% among antidepressant non-users with a mean age of 38.26 ± 12.9 (SD). Post-PSM, cohorts were well-matched (Supplementary Data 3).

The antidepressant users had a significantly higher risk for mortality than the matched non-users (1.6% vs. 1.01%), with a Cox proportional hazard ratio (HR) of 1.35 (95% CI 1.26 to 1.45). Consistently higher risks were observed in patients stratified by sex, age group, BMI, and racial and ethnic classifications (Fig. 2a, b). SSRI users showed a higher mortality rate than non-users (HR, 1.32 [CI, 1.22 to 1.44]) (Fig. 2a, Supplementary Data 4). Similarly, TCA (2.02% vs. 1.27%) (HR, 1.27 [CI, 1.14 to 1.43]) (Fig. 2a, Supplementary Data 5) and SNRIs users (2.23% vs. 1.47%) (HR, 1.32 [CI, 1.19 to 1.46]) (Fig. 2a, Supplementary Data 6) experienced an increased mortality rate. A pronounced difference was observed among mirtazapine users, with the mortality rate reaching 5.29% compared to 2.27% in non-users (HR, 2.05 [CI, 1.73 to 2.43]) (Fig. 2a, Supplementary Data 7).

**All-cause mortality associated with antispasmodic use.** Among antispasmodic users, 73.7% were female with a mean age of 37.6 ± 13.26 (SD), compared to 69.6% among non-users with a mean age of 37.36 ± 13.06 (SD). Post-PSM, cohorts were well-matched for demographics, comorbidities, and medication use (Supplementary Data 8).

Antispasmodic users did not have a higher mortality rate than non-users (1.44% vs. 1.60%; HR, 0.95 [CI, 0.89 to 1.00]). Mortality risk did not increase when stratified by hyoscyamine (1.46% vs. 1.78%; HR, 0.83 [CI, 0.76 to 0.91]) or dicyclomine (1.6% vs. 1.79%; HR, 0.99 [CI, 0.94 to 1.06]). Furthermore, mortality risk was statistically similar across stratifications by sex, age group, BMI, ethnicity, and race (Fig. 2c, d).

## Specification of target trials and active comparator
A comparative analysis was conducted between antidepressant users, including all subgroups, and antispasmodic users, with the latter serving as the active comparator group. The comparator cohort consisted of patients prescribed antispasmodics without prior prescriptions for antidepressants. Kaplan–Meier survival curves demonstrated a clear divergence between antidepressant and antispasmodic users after five years, highlighting the long-term association of antidepressant use on mortality (Fig. 3 and Supplementary Data 9).

## Medication classes used in IBS subtypes
**All-cause mortality in patients with IBS-D.** All-cause mortality rates were not elevated in patients taking cholestyramine/colestipol (1.05% vs. 1.31%; HR, 0.95 [CI, 0.56 to 1.61]), or eluxadoline (1.16% vs. 1.16%; HR, 0.63 [CI, 0.18 to 2.18]), rifaximin, or antispasmodics, (0.68% vs. 0.79%; HR, 1.11 [CI, 0.74 to 1.65]), and (0.58% vs. 0.78%; HR, 0.79 [CI, 0.72 to 1.31]), respectively. Significantly elevated mortality risks were seen in patients taking antidepressants (0.77% vs. 0.57%; HR, 1.50 [CI 1.07 to 2.11]), diphenoxylate (2.34% vs. 1.38%; HR, 1.89 [CI 1.02 to 3.51]), and loperamide 2.1% vs. 0.98%; HR, 2.39 [CI, 1.48 to 3.90] (Fig. 4a).

A comparative analysis was performed between patients with IBS-D receiving mu receptor agonists and those receiving antispasmodics as the active comparator group. Kaplan–Meier survival analyses revealed a distinct divergence in survival probabilities between mu receptor agonist users and antispasmodic users after three years, indicating long-term association with mortality (Supplementary Fig. 1).

**All-cause mortality in patients with IBS-C.** All-cause mortality rates were not significantly increased in patients using PEG-3350 (0.49% vs. 0.35%; HR, 1.53 [CI, 0.80 to 2.65]), secretagogues (1.39% vs. 1.39%; HR, 2.55 [CI, 6.28 to 10.13]), or antispasmodics (0.46% vs. 0.49%; HR, 1.02 [CI, 0.58 to 1.79]). Patients with IBS-C who were administered antidepressants demonstrated a significantly increased mortality risk (0.82% vs. 0.64%, HR, 1.56 [CI, 1.08 to 2.25]). (Fig. 4b).

**Secondary outcome.** Patients with IBS who were prescribed antidepressants experienced a significantly higher incidence of secondary outcomes, including hypertension, cardiac arrhythmia, inpatient encounters, heart failure, ischemic heart disease, aspiration pneumonitis, falls, stroke, serotonin syndrome, suicidal ideation, gastrointestinal bleeding, and obesity. The hazard ratios for these conditions ranged from 1.21 to 5.19, indicating an increased risk compared to non-users. (Supplementary Fig. 2).

The association between antidepressant prescription refills and mortality in patients with IBS was investigated across various refill counts (2, 4, 8, 12, 16, and 20 refills). An increased risk of mortality was observed with higher refill numbers compared to non-users, with the highest risk occurring at 20 refills (HR, 1.91 [CI, 1.68 to 2.17]) (Supplementary Fig. 3).

**Negative control outcomes.** Analyses of negative control outcomes, including blepharitis, frostbite, acute appendicitis, nummular dermatitis, eczematous dermatitis of the eyelid, retinal detachments, and protozoal diseases, showed no significant event rates (Supplementary Fig. 4).

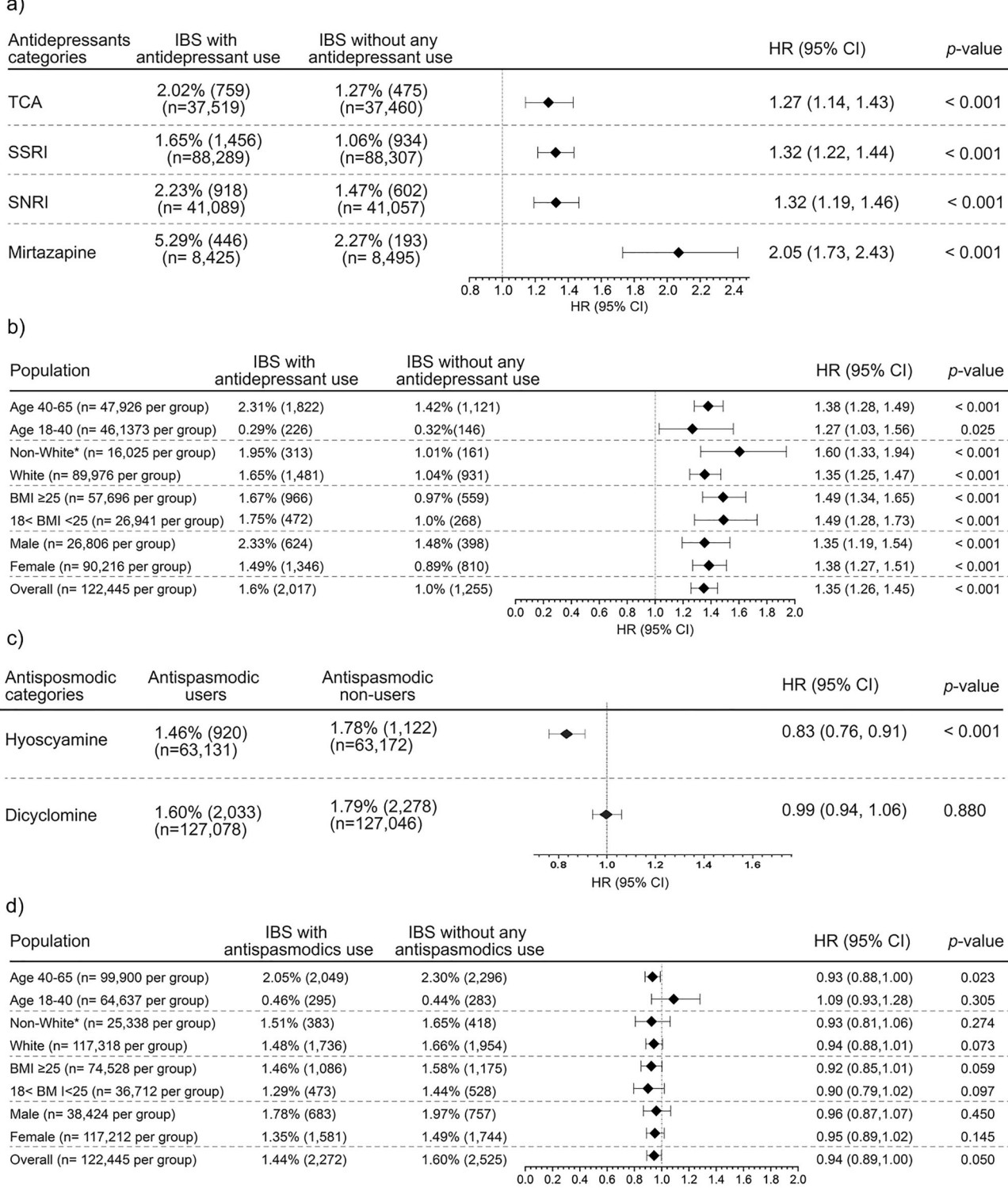

**Fig. 2 | Comparison of all-cause mortality risk and hazard ratios. a** Among patients with IBS using TCA, SSRI, SNRI, and mirtazapine versus non-users. **b** Subgroup analysis by age, race, BMI, and sex among antidepressant users and non-users. **c** Among patients with IBS using hyoscyamine or dicyclomine versus non-users, and **d** subgroup analysis by age, race, BMI, and gender among antispasmodic users and non-users. Propensity score matching (PSM) was applied to balance baseline characteristics. Data are shown as HRs with 95% CIs; markers indicate HRs and bars the 95% CIs. HRs were derived from Cox proportional hazards models using two-sided Wald tests incorporating daily time intervals and censoring to evaluate the probability of outcomes. No multiple comparison adjustment was applied. Exact *p*-values are reported when available; TriNetX displays *p* < 0.001 for values below this threshold. BMI body mass index, HR Hazard ratio, CI Confidence interval. TCA tricyclic antidepressant, SSRI selective serotonin reuptake inhibitor, SNRI serotonin-norepinephrine reuptake inhibitor, BMI body mass index, HR Hazard ratio, CI Confidence interval. *Non-White: American Indian or Alaska Native, Asian, Black or African American, Native Hawaiian or Other Pacific Islander, Other Race

**Fig. 3 | Survival probability curves for patients with IBS comparing antispasmodic users with antidepressant users, SSRI users, TCA users, SNRI users, and mirtazapine users, up to a 15-year follow-up period in target trial emulations.** Solid lines depict survival probability over time, and shaded regions indicate 95% confidence intervals. TCA tricyclic antidepressant, SSRI selective serotonin reuptake inhibitor, SNRI serotonin-norepinephrine reuptake inhibitor, Antispasmodics: hyoscyamine, dicyclomine

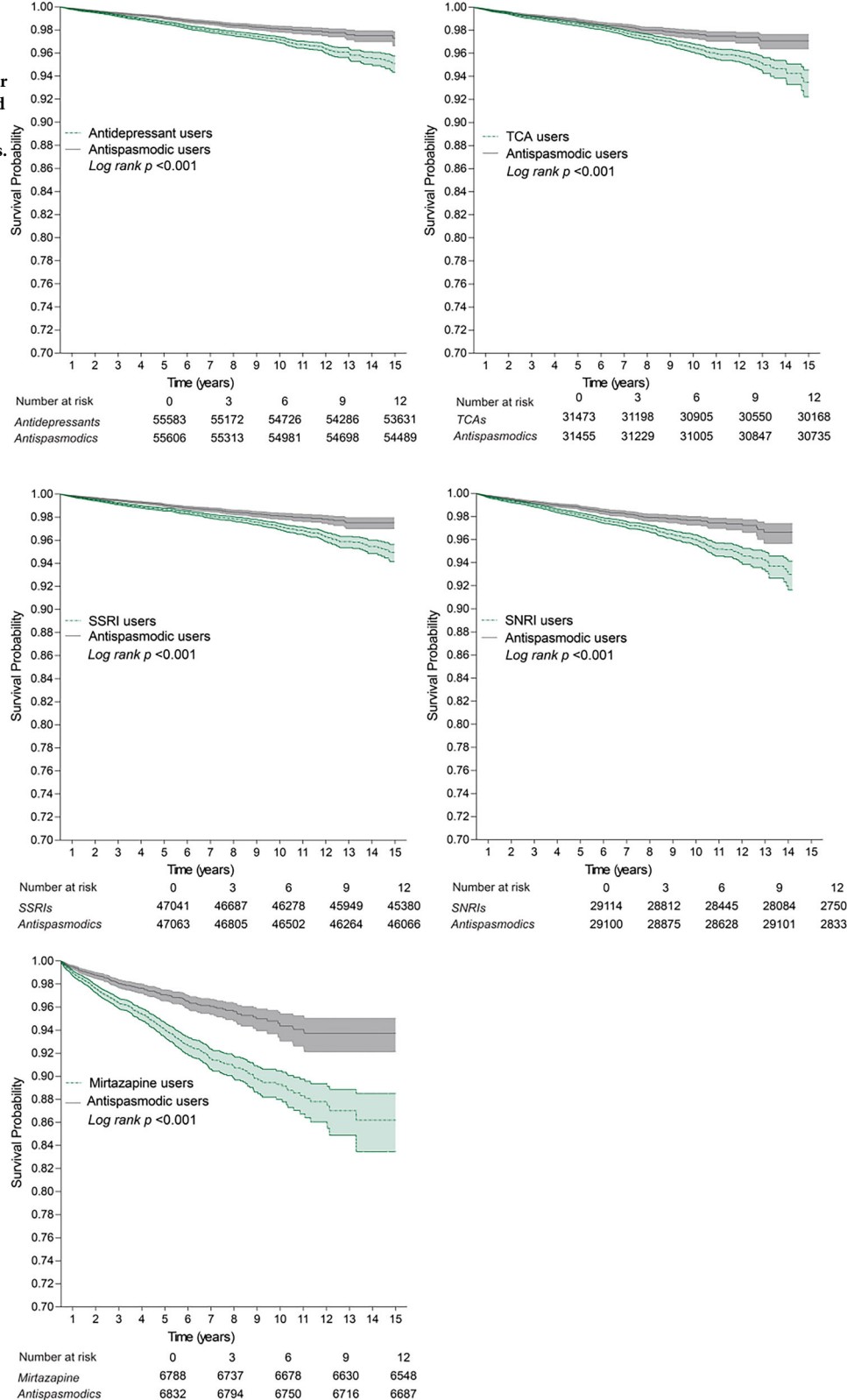

## Discussion

This study provides a comprehensive real-world PSM assessment of the association between recommended pharmacotherapies for IBS and all-cause mortality. Across all IBS subtypes, the use of antidepressants was consistently linked to a significantly higher risk of mortality compared to non-users, whereas antispasmodic use did not increase mortality risk in these patients. Patients with IBS-D using loperamide, diphenoxylate, and antidepressants had an increased risk of mortality. However, no significant increase was observed in patients treated with antispasmodics (dicyclomine and hyoscyamine), eluxadoline, cholestyramine, colestipol, or rifaximin. In IBS-C patients, the use of recommended medications such as PEG-3350, antispasmodics, and secretagogues was not associated with an elevated

a)

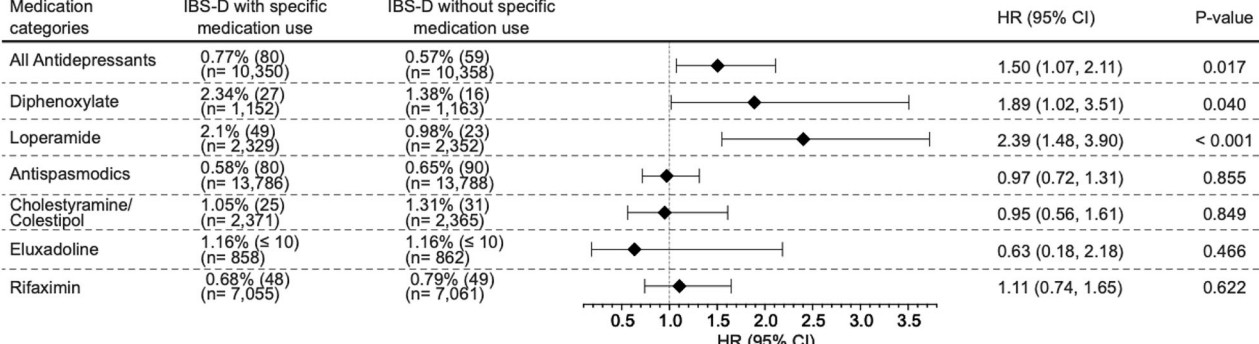

| Medication categories | IBS-D with specific medication use | IBS-D without specific medication use | | HR (95% CI) | P-value |
|---|---|---|---|---|---|
| All Antidepressants | 0.77% (80) (n= 10,350) | 0.57% (59) (n= 10,358) | | 1.50 (1.07, 2.11) | 0.017 |
| Diphenoxylate | 2.34% (27) (n= 1,152) | 1.38% (16) (n= 1,163) | | 1.89 (1.02, 3.51) | 0.040 |
| Loperamide | 2.1% (49) (n=2,329) | 0.98% (23) (n= 2,352) | | 2.39 (1.48, 3.90) | < 0.001 |
| Antispasmodics | 0.58% (80) (n= 13,786) | 0.65% (90) (n= 13,788) | | 0.97 (0.72, 1.31) | 0.855 |
| Cholestyramine/ Colestipol | 1.05% (25) (n= 2,371) | 1.31% (31) (n= 2,365) | | 0.95 (0.56, 1.61) | 0.849 |
| Eluxadoline | 1.16% (≤ 10) (n= 858) | 1.16% (≤ 10) (n= 862) | | 0.63 (0.18, 2.18) | 0.466 |
| Rifaximin | 0.68% (48) (n= 7,055) | 0.79% (49) (n= 7,061) | | 1.11 (0.74, 1.65) | 0.622 |

b)

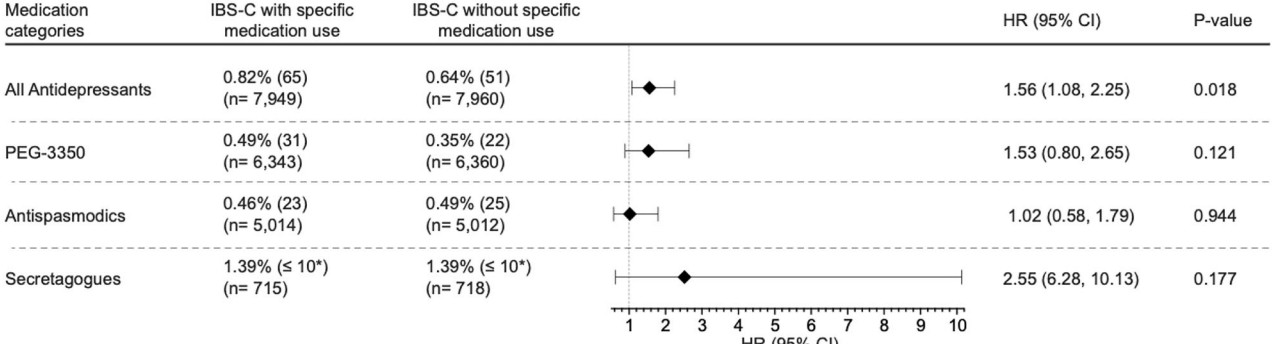

| Medication categories | IBS-C with specific medication use | IBS-C without specific medication use | | HR (95% CI) | P-value |
|---|---|---|---|---|---|
| All Antidepressants | 0.82% (65) (n= 7,949) | 0.64% (51) (n= 7,960) | | 1.56 (1.08, 2.25) | 0.018 |
| PEG-3350 | 0.49% (31) (n= 6,343) | 0.35% (22) (n= 6,360) | | 1.53 (0.80, 2.65) | 0.121 |
| Antispasmodics | 0.46% (23) (n= 5,014) | 0.49% (25) (n= 5,012) | | 1.02 (0.58, 1.79) | 0.944 |
| Secretagogues | 1.39% (≤ 10*) (n= 715) | 1.39% (≤ 10*) (n= 718) | | 2.55 (6.28, 10.13) | 0.177 |

**Fig. 4 | Comparison of all-cause mortality risk and hazard rates. a** Comparison of all-cause mortality risk and hazard rates among patients with IBS-D who were prescribed IBS-D-recommended medications versus non-users. **b** Comparison of all-cause mortality risk and hazard rates among patients with IBS-C who were prescribed IBS-C-recommended medications versus non-users. Propensity score matching (PSM) was utilized to balance baseline characteristics between users and non-users. Data are shown as HRs with 95% CIs; markers indicate HRs and bars the 95% CIs. HRs were derived from Cox proportional hazards models using two-sided Wald tests incorporating daily time intervals and censoring to evaluate the probability of outcomes. No multiple comparison adjustment was applied. Exact p-values are reported when available; TriNetX displays p < 0.001 for values below this threshold probability of outcomes. HR Hazard ratio, CI Confidence interval. Secretagogues: tenapanor, plecanatide, lubiprostone, linaclotide; Antispasmodics: hyoscyamine, dicyclomine. *To protect patient privacy, numbers are rounded up to 10. This may impact results, particularly for small cohorts and infrequent outcomes.

mortality risk. However, similar to IBS-D, antidepressants were associated with an increased mortality rate in patients with IBS-C.

The increased risk of mortality associated with antidepressant users remained consistent across subgroups stratified by sex, age, BMI, and ethnicity, as well as specific antidepressant classes, including SSRIs, SNRIs, TCAs, and mirtazapine. Our subsequent findings suggest that the associations observed in the primary analysis are unlikely to be influenced by unmeasured confounding or systematic bias. First, a greater number of antidepressant refills, ranging from 2 to 20 during the follow-up period, was linked to a progressively higher relative risk of mortality. Second, when comparing all or specific classes of antidepressants to an active comparator (antispasmodics), the mortality rate consistently remained elevated in the antidepressant group. Third, analyses of negative control outcomes showed no significant associations with all-cause mortality.

Emerging evidence suggests that the long-term use of antidepressants may be associated with an increased risk of adverse health outcomes, including elevated mortality rates. Antidepressants may increase arrhythmia risk, particularly through QT prolongation, which is linked to sudden cardiac events[24-26]. Hypertension is another concern, as increased serotonergic activity may raise blood pressure[27]. Excess serotonergic stimulation raises the risk of serotonin syndrome, a life-threatening condition[28]. TCAs have been associated with a 28% increased risk of stroke[29]. Antidepressants may also contribute to respiratory complications[30], including a higher prevalence of aspiration pneumonitis and inpatient encounters, particularly among patients with comorbidities[31]. Additionally,

antidepressant use, especially SSRIs, has been linked to a higher risk of falls[32]. Another critical concern is the association between antidepressant use and suicidality, particularly in younger populations[33]. The antiplatelet properties of SSRIs may contribute to an increased risk of gastrointestinal bleeding[34]. Antidepressant use is also strongly linked to weight gain[35].

Aligned with the literature, we observed a 21% increase in inpatient visits (p < 0.0001), a 35% increase in ischemic heart disease or heart failure events, and a 43% increase in hypertension with antidepressants. Additionally, antidepressant use was associated with a 70% higher risk of aspiration pneumonia, cerebral infarction, and fall-related incidents, a 60% increase in serotonin syndrome, a 47% increase in gastrointestinal bleeding, an 80% higher risk of obesity, and a fivefold increase in suicidal ideation.

Approximately 20% of the US gastroenterologists and primary care providers "most of the time" or "always" recommend antidepressants for IBS-D and IBS-C[9]. Our study underscores the need for caution when prescribing antidepressants for IBS and highlights the importance of further research to elucidate underlying mechanisms and guide safer, evidence-based treatment strategies.

Loperamide, a μ-opioid receptor agonist[36], was associated with a 2.39-fold increased risk of all-cause mortality in IBS-D. Loperamide typically lacks central opioid effects at recommended doses (2–8 mg/day) due to low oral bioavailability and P-glycoprotein efflux[37]; however, excessive use can lead to severe cardiac dysrhythmias[38]. Loperamide is suspected to block sodium and potassium channels in the myocardium, leading to QT prolongation, QRS widening, Torsades de Pointes, and other ventricular

arrhythmias[39], which may progress to cardiac arrest[40]. Because of the low number of incident mortality cases, we were unable to explore the underlying causes in greater detail. Increased risk of mortality was also seen in patients with IBS-D taking diphenoxylate/atropine (HR = 1.89), which is a combined mu and muscarinic receptor agonist[41].

Other recommended treatments for IBS-D based on guidelines, including cholestyramine/colestipol, eluxadoline, and rifaximin, were not associated with an increased risk of mortality. Similarly, for IBS-C, PEG-3350 and secretagogues showed no significant risk. Noticeably, medications that were not associated with an increased risk of mortality include all the FDA-approved medications: lubiprostone, linactolide, plecanatide, tenanpanor, eluxadoline, and rifaximin. The FDA rigorously evaluates IBS treatments, but typically only for trials under a year. Reassuringly, our data offers valuable long-term safety insights. Currently, fewer than 20% of IBS-C and less than 10% of IBS-D patients report ever having been treated with FDA-approved medications[9].

Our study has several strengths, including long-term follow-up of a large nationwide cohort spanning outpatient and inpatient settings, as well as federally and commercially insured patients. Additionally, we achieved balanced PSM for 56 potential confounders. A key strength of our analysis is the use of Cox proportional hazards modeling, which enables the evaluation of time-to-event outcomes while appropriately handling censored data and adjusting for variable follow-up durations. We also employed a trial-emulation active-comparator design, comparing antispasmodics with different groups of antidepressants to overcome potential prevalent user bias and to reduce susceptibility to confounding by indication[42]. We incorporated a time lag between drug exposure and outcome measurement to address concerns related to reverse causality, disease latency, and to minimize the risk of immortal time bias[43]. Moreover, we included several negative control outcomes, which served as robustness checks for the associations of interest.

This retrospective observational study, which utilized patient EHRs, is subject to several inherent limitations, including potential for overdiagnosis, underdiagnosis, misdiagnosis, and the presence of unmeasured/uncontrolled confounding variables. As a result, some findings may reflect residual confounding due to factors not accounted for during the PSM process. The reliance on claims data rather than medical records may introduce misclassification errors. These errors are unlikely to differ between treatment groups, which would bias the estimates toward the null value. Therefore, the observed differences between groups might underestimate the true magnitude of effect. Similarly, nondifferential misclassification of binary exposures and outcomes, as seen in claims data-based studies, tends to bias effect estimates toward the null[44]. The study used data from TriNetX, requiring validation with other databases for generalizability. Due to the constraints of the TriNetX platform, we were unable to perform multiple rounds of PSM or conduct competing risk analyses, which could have improved model precision and reduced bias. The platform does not allow for identifying individuals who missed prescription refills. To address this, we conducted a sensitivity analysis (2–20 refills), showing higher refills correlated with increased mortality. Lastly, while we provided all-cause mortality estimates to assess competing risks of death, limitations of the TriNetX platform prevented us from determining cause-specific mortality. Future studies with access to more granular data and advanced analytical methods should aim to address this gap.

In conclusion, our results support concerns about an increased risk of all-cause mortality with antidepressants in IBS and mu receptor agonists in IBS-D. Other commonly recommended pharmacotherapies for IBS do not appear to be associated with increased mortality risk. Our analyses can help inform future societal guidelines and contribute to the development of algorithmic approaches to IBS pharmacotherapy, leveraging long-term safety data.

## Data availability

The data supporting this study are derived from the TriNetX Research Network. Due to data use agreements and patient privacy protections, individual-level data are not publicly available. Access to the data is subject to TriNetX approval.

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

## Author contributions

A.R. conceived the study. A.R. and S.M. designed the study. S.M., P.N., and N.B. performed the data analysis and created the tables and figures. A.R., W.H, M.P., W.T., Y.H.Y., P.N., N.B., and S.M. interpreted the results and drafted the paper. S.M., A.R., W.H, M.P., and W.T. critically contributed to study design, result interpretation, and paper preparation. A.R. and S.M. had full access to all the data in the study and take responsibility for the integrity of the data and the accuracy of data analysis.

## Competing interests

These authors disclose the following: Mark Pimentel is also a consultant for and received grant support from Bausch Health. Ali Rezaie reports serving as a consultant for Bausch Health and Ardelyx. In addition, Cedars-Sinai Medical Center has a licensing agreement with Gemelli Biotech. Ali Rezaie and Mark Pimentel have equity in Gemelli Biotech and Good LFE. The remaining authors disclose no conflicts.
