## [Transparent Peer Review file · Communications Medicine]

Association of pharmacotherapy with all-cause mortality among patients with irritable bowel syndrome

Corresponding Author: Dr Ali Rezaie

Version 0:

Reviewer comments:

Reviewer #1

(Remarks to the Author)

Introduction

Rationale for the study: IBS is a benign but frequently invalidating condition that significantly affects QOL. Medication is one of the cornerstones (together with diet, life style and psychological interventions) in the treatment of IBS symptoms. Some drugs are FDA and/or EMA approved, for some drugs evidence exist with respect to therapeutic efficacy. Several other drugs used in IBS have not been approved and/or lack evidence with respect to efficacy. IBS is a chronic condition and medication use is often chronic or for longer periods. Medication in IBS should be safe and possible without or with minimum side effects or risks.

The rationale for this study, to assess all-cause mortality in a large population-based IBS cohort related to use of recommended IBS medication classes, is relevant.

Methods

The study data were obtained from various healthcare organizations, primarily large academic medical centers spanning all 50 US states. IBS is a prevalent disorder and most patients do not visit a doctor or are treated in primary care. The more severe cases are referred to a specialist and treated in secondary or tertiary care.

The present cohort may not reflect IBS among the population but among IBS patients treated in secondary or even tertiary care. This should be discussed as it may lead to selection bias.

The use of propensity score-matched (PSM) cohort is adequate.

The consequences of the use of claims data instead of medical diagnoses data should be discussed. How reliable are the co-morbid "medical diagnoses" mentioned in tables and supplementary data? (validity of data)

Results

The fact that antidepressants (according to guide-lines considered as second line treatment) were the most commonly used medication (over 50% of prescribed drugs) in this cohort, again shows that this large cohort is composed of "more difficult" or "severe" IBS cases (selection bias).

Antidepressants may have been prescribed for typical IBS symptoms (pain, discomfort) but also for comorbid symptoms of anxiety and depression. Please comment.

page 11, line 210: mortality risks in IBS-D patients taking antidepressants: 0.77% vs 57% (??) HR 1.50; should this be 0.57%?

Seen from (my) clinical perspective: psychological/psychiatric comorbidity and their related symptoms are highly prevalent among IBS patients, which is logical as IBS is considered to be a disorder of gut-brain interaction (DGBI). Published prevalence of symptoms of anxiety and depression in IBS is high (resp. around 40% and 30%) and of anxiety and depression disorders also (resp. 23% and 23%) (ref: Zamani M et al. Systematic review with meta-analysis: the prevalence

of anxiety and depression in patients with irritable bowel syndrome. *Aliment. Pharmacol. Ther.* 2019 Jul;50(2):132-143). Thus, symptoms and disorders of anxiety and depression are highly prevalent among IBS patients but largely go unrecognized unless specifically searched for. A large discrepancy is noticed among (low prevalence) of psychiatric diagnoses in the IBS cohort vs reported psychiatric diagnoses in IBS in the literature. Please comment.

Patients having an anxiety or depressive disorder are known to have a higher mortality rate (extensive literature). I am curious to know the impact of a psychiatric diagnosis as factor to the higher mortality rate observed in the antidepressants users. How reliable are the diagnoses based on claim instead of based on medical records?

Causal pathways between mental disorders and mortality include various factors: direct effects (for instance suicide, accidents), adverse life style factors (alcohol consumption smoking, physical inactivity, diet) and also drug use (especially antipsychotic drugs and antidepressants).

The authors have taken into account psychiatric diagnoses in IBS (but these are probably underreported or less reliable) and in fact many IBS patients may have had underlying psychiatric/psychological disorder.

Be careful with interpretation of findings: the study is on associations and not on causality.

I fully agree with the need for caution with respect to prescribing antidepressants in IBS. This drug class should not be considered as first line medical therapy in IBS but as a second line (as noted in many guidelines and studies (see article and editorial on the role of neuromodulators in IBS: *Lancet Gastroenterology Hepatology* June 2025)). This could be added to the discussion.

Reviewer #2

(Remarks to the Author)

The manuscript reports that association of pharmacotherapy with all-cause mortality among patients with irritable bowel syndrome. In this manuscript, a nationwide US electronic health record database was used to assess the association between long-term pharmacotherapies and all-cause mortality in patients with IBS and its subtype. They concluded that antidepressant use was associated with an increased risk of all-cause mortality. This association remained consistent across antidepressant subclasses and demographic subgroups. Other pharmacotherapies including antispasmodic use did not show any significant associations with all-cause mortality. There are some major concerns on this manuscript, maybe due to their retrospective design and limited access to the healthcare record to categorize the IBS severity or other important morbid conditions.

1. One major concern regarding the manuscript is the potential impact of confounding factors. The table highlights differences in the prevalence of several conditions—including psychological disorders, cardiovascular disease, kidney problems, and diabetes—all of which are known to be associated with increased mortality risk. Notably, many of these conditions were more common among IBS patients using antidepressants, which could significantly influence the study's findings. Were these confounding factors accounted for in the analysis of mortality risk?
2. Severity of IBS symptoms: Antidepressants are more likely to be prescribed to patients with more severe forms of IBS, particularly when other pharmacological treatments have failed. Similar to the first concern, this may act as an additional confounding factor that could bias the results or lead to alternative interpretation.
3. Negative control outcomes: A more detailed explanation of the negative control outcomes would be helpful. Were these conditions compared between patients using antidepressants and those not using them? Additionally, the manuscript refers to supplemental Table 4d—could you clarify its location or provide it if missing?

Version 1:

Reviewer comments:

Reviewer #1

(Remarks to the Author)

The questions and points I raised have been adequately answered. The answers have convinced me of the thoroughness of the author's approach, methodology and interpretation of data. Pitfalls have been recognized, conclusions are in line with findings and data.

Reviewer #2

(Remarks to the Author)

The responses for comments are well prepared. No major concerns.

Referee #1: GI, clinician, epidemiology

Referee #2: IBS, epidemiology

Reviewers' comments:

Reviewer #1 (Remarks to the Author):

Introduction

Rationale for the study: IBS is a benign but frequently invalidating condition that significantly affects QOL. Medication is one of the cornerstones (together with diet, life style and psychological interventions) in the treatment of IBS symptoms. Some drugs are FDA and/or EMA approved, for some drugs evidence exist with respect to therapeutic efficacy. Several other drugs used in IBS have not been approved and/or lack evidence with respect to efficacy. IBS is a chronic condition and medication use is often chronic or for longer periods. Medication in IBS should be safe and possible without or with minimum side effects or risks. The rationale for this study, to assess all-cause mortality in a large population-based IBS cohort related to use of recommended IBS medication classes, is relevant.

Thank you for your thoughtful comment and for recognizing the relevance of our study rationale in the context of IBS management and medication safety. We truly appreciate your supportive feedback.

Methods

The study data were obtained from various healthcare organizations, primarily large academic medical centers spanning all 50 US states. IBS is a prevalent disorder and most patients do not visit a doctor or are treated in primary care. The more severe cases are referred to a specialist and treated in secondary or tertiary care.

The present cohort may not reflect IBS among the population but among IBS patients treated in secondary or even tertiary care.

Q1) This should be discussed as it may lead to selection bias.

Response: We appreciate the reviewer's insightful comment. We agree that the use of an administrative database such as TriNetX may introduce selection bias, as patients captured are more likely to represent those with moderate to severe presentations managed in secondary or tertiary care settings. Nonetheless, a key strength of the TriNetX network is its inclusion of data from diverse healthcare organizations, including a large proportion of outpatient encounters, which broadens generalizability compared to databases limited to hospitalized populations. We acknowledge that such selection bias is an inherent limitation of administrative databases, and this has been clearly noted in our manuscript.

The use of propensity score-matched (PSM) cohort is adequate.

Q2) The consequences of the use of claims data instead of medical diagnoses data should be discussed. How reliable are the co-morbid "medical diagnoses" mentioned in tables and supplementary data? (validity of data)

Response: Thank you for raising this important point. We fully agree that our use of an administrative or claims based data source rather than clinically adjudicated diagnosis

records may affect the validity of the comorbid “medical diagnoses” reported in our tables and supplementary data. Diagnostic coding using ICD and similar administrative codes does not perfectly correspond to a clinician verified diagnosis, and prior work has shown variation in accuracy depending on the condition and coding algorithm. For example, in our previous work in a Canadian administrative database study of Inflammatory Bowel Disease (IBD), case definitions that required multiple claims, hospitalizations, and ambulatory contacts reached high specificity (>99%) but only moderate sensitivity (~80%) when compared to endoscopy or chart review as the gold standard (PMC3472911).

In the present study, we used specific ICD codes for diseases which are expected to minimize misclassification. Importantly, any residual misclassification is likely to be nondifferential between exposure groups, which would bias estimates toward the null rather than inflate associations

Finally, we explicitly acknowledged this limitation in the *Discussion* and emphasized that while claims data are not equivalent to adjudicated diagnoses, they offer the advantage of very large sample sizes and long-term follow-up, enabling meaningful population-level inference

Results

Q3) The fact that antidepressants (according to guide-lines considered as second line treatment) were the most commonly used medication (over 50% of prescribed drugs) in this cohort, again shows that this large cohort is composed of “more difficult” or “severe” IBS cases (selection bias).

Response: We agree with the reviewer that the predominance of antidepressant prescriptions, which are generally considered second-line therapies in IBS management, likely reflects inclusion of patients with more severe or refractory disease. As noted, this pattern supports the possibility of selection bias, since individuals treated at secondary or tertiary care centers are more likely to have complex or treatment-resistant IBS. We have acknowledged this limitation in the Discussion and clarified that our cohort may represent a subset of patients with more severe symptom burden rather than the general IBS population.

Q4) Antidepressants may have been prescribed for typical IBS symptoms (pain, discomfort) but also for comorbid symptoms of anxiety and depression. Please comment.

Response: We agree with the reviewer that antidepressants may be prescribed not only for IBS-related symptoms such as pain and discomfort, but also for comorbid psychiatric conditions including anxiety and depression. To account for this potential confounding, we performed 1:1 propensity score matching (PSM) between treatment groups, incorporating baseline psychiatric comorbidities and other relevant covariates in the model. The matching achieved excellent balance across variables, with all standardized mean differences (SMDs) below 0.1, indicating minimal residual confounding.

Importantly, these findings remained consistent when antidepressant users were compared to an active comparator group (antispasmodic users), which further strengthens our confidence that residual confounding is unlikely to fully explain the observed associations. Moreover, negative control outcomes were included to assess potential residual bias or spurious associations in our analysis.

While residual confounding can never be fully eliminated in observational studies, this analytic approach substantially mitigates bias arising from differential antidepressant prescribing for psychiatric indications.

Q5) page 11, line 210: mortality risks in IBS-D patients taking antidepressants: 0.77% vs 57% (??) HR 1.50; should this be 0.57%?

Response: Thank you for bringing this to our attention. We appreciate the careful review and have corrected the typographical error in the manuscript. The correct value is 0.57%, not 57%.

Seen from (my) clinical perspective: psychological/psychiatric comorbidity and their related symptoms are highly prevalent among IBS patients, which is logical as IBS is considered to be a disorder of gut-brain interaction (DGBI). Published prevalence of symptoms of anxiety and depression in IBS is high (resp. around 40% and 30%) and of anxiety and depression disorders also (resp. 23% and 23%) (ref: Zamani M et al. Systematic review with meta-analysis: the prevalence of anxiety and depression in patients with irritable bowel syndrome. *Aliment. Pharmacol. Ther.* 2019 Jul;50(2):132-143). Thus, symptoms and disorders of anxiety and depression are highly prevalent among IBS patients but largely go unrecognized unless specifically searched for.

Q6) A large discrepancy is noticed among (low prevalence) of psychiatric diagnoses in the IBS cohort vs reported psychiatric diagnoses in IBS in the literature. Please comment.

Response: We thank the reviewer for this insightful comment. We agree that psychological and psychiatric comorbidities are highly prevalent among IBS patients, consistent with the concept of IBS as a disorder of gut-brain interaction. Our cohort demonstrated comparable rates to those reported in the literature, with 52% of patients having anxiety or stress-related disorders (F40-F48) and 39% having mood [affective] disorders (F30-F39), which aligns closely with the prevalence estimates reported by Zamani et al. (*Aliment Pharmacol Ther* 2019). This similarity supports the validity of our dataset in capturing psychiatric comorbidities among IBS patients.

To mitigate potential confounding from psychiatric comorbidities that were captured, we included all major psychiatric diagnostic categories in our 1:1 propensity score matching (PSM) model. Matching achieved excellent balance (all standardized mean differences < 0.1), ensuring that residual confounding from recorded psychiatric conditions was minimized, although we acknowledge that unmeasured subclinical symptoms may still exist.

These detailed psychiatric data were not included in the manuscript but were derived from the underlying TriNetX analysis.

Q7) Patients having an anxiety or depressive disorder are known to have a higher mortality rate (extensive literature). I am curious to know the impact of a psychiatric diagnosis as factor to the higher mortality rate observed in the antidepressants users. How reliable are the diagnoses based on claim instead of based on medical records?

Causal pathways between mental disorders and mortality include various factors: direct effects (for instance suicide, accidents), adverse life style factors (alcohol consumption smoking, physical inactivity, diet) and also drug use (especially antipsychotic drugs and antidepressants).

The authors have taken into account psychiatric diagnoses in IBS (but these are probably underreported or less reliable) and in fact many IBS patients may have had underlying psychiatric/psychological disorder.

Response: We appreciate the reviewer's insightful comment. Indeed, patients with anxiety or depressive disorders are known to have higher mortality risk, as supported by extensive literature. The causal pathways between psychiatric disorders and mortality are multifactorial and include direct effects (such as suicide and accidental death), adverse lifestyle factors (e.g., alcohol use, smoking, physical inactivity, poor diet), and medication-related effects (particularly from antipsychotic and antidepressant use).

We acknowledge that in our study, psychiatric diagnoses were identified using ICD codes from an administrative database, which may underestimate or misclassify certain conditions compared to diagnoses derived from detailed medical records. However, this limitation applies equally to both the antidepressant and control cohorts, minimizing differential bias. To further account for this, we included psychiatric comorbidities in our propensity score matching model, which achieved good balance (standardized mean differences <0.1), thereby reducing confounding by measured psychiatric factors.

While TriNetX does not provide specific causes of death, we performed secondary outcome analyses using surrogate clinical endpoints to explore potential contributors to mortality. Nevertheless, the lack of cause specific mortality data remains an inherent limitation of the database and should be explored in other databases where such data is available.

Q8) Be careful with interpretation of findings: the study is on associations and not on causality.

Response: We appreciate the reviewer's important reminder. We have been very careful in our interpretation of the results and have consistently emphasized that this study identifies associations rather than causality. Our findings should therefore be interpreted as observational correlations, not as evidence of a direct causal relationship.

I fully agree with the need for caution with respect to prescribing antidepressants in IBS. This

drug class should not be considered as first line medical therapy in IBS but as a second line (as noted in many guidelines and studies)

Q7 (see article and editorial on the role of neuromodulators in IBS: Lancet Gastroenterology Hepatology June 2025). This could be added to the discussion.

Response: We fully agree with the reviewer regarding the need for caution in prescribing antidepressants for IBS. As highlighted in multiple guidelines and recent publications (including the *Lancet Gastroenterology & Hepatology*, June 2025, article and editorial on the role of neuromodulators in IBS), antidepressants should not be considered first-line therapy but rather as a second-line option for patients with persistent or refractory symptoms. We have clarified this point in the Discussion, noting that while our study examined associations with antidepressant use, the findings do not support their routine use as primary therapy. Furthermore, longer-term outcome data and studies focused on FDA-approved IBS medications are needed to better define the optimal treatment strategy.

Reviewer #2 (Remarks to the Author):

The manuscript reports that association of pharmacotherapy with all-cause mortality among patients with irritable bowel syndrome. In this manuscript, a nationwide US electronic health record database was used to assess the association between long-term pharmacotherapies and all-cause mortality in patients with IBS and its subtype. They concluded that antidepressant use was associated with an increased risk of all-cause mortality. This association remained consistent across antidepressant subclasses and demographic subgroups. Other pharmacotherapies including antispasmodic use did not show any significant associations with all-cause mortality. There are some major concerns on this manuscript, maybe due to their retrospective design and limited access to the healthcare record to categorize the IBS severity or other important morbid conditions.

1. One major concern regarding the manuscript is the potential impact of confounding factors. The table highlights differences in the prevalence of several conditions—including psychological disorders, cardiovascular disease, kidney problems, and diabetes—all of which are known to be associated with increased mortality risk. Notably, many of these conditions were more common among IBS patients using antidepressants, which could significantly influence the study's findings. Were these confounding factors accounted for in the analysis of mortality risk?

Response: We appreciate the reviewer's thoughtful comment and fully acknowledge the importance of accounting for potential confounding factors. We performed a meticulous analysis to minimize their impact on the observed associations. Specifically, we conducted 1:1 propensity score matching (PSM) incorporating all major demographic and clinical variables, including psychological disorders, cardiovascular disease, chronic kidney disease, diabetes, and other comorbidities known to influence mortality risk. The matching achieved excellent balance across all covariates (standardized mean differences <0.1), indicating that the groups were well matched.

In addition, we used an active comparator design, comparing antidepressant users with IBS patients treated with other commonly used medications, to further reduce confounding by indication. These methodological steps strengthen the robustness of our findings, though we acknowledge that residual confounding cannot be completely excluded in observational studies.

We have further elaborated on the limitations of our study in the discussion.

2. Severity of IBS symptoms: Antidepressants are more likely to be prescribed to patients with more severe forms of IBS, particularly when other pharmacological treatments have failed. Similar to the first concern, this may act as an additional confounding factor that could bias the results or lead to alternative interpretation.

Response: We agree with the reviewer that antidepressants are more often prescribed to patients with more severe or refractory IBS symptoms, which could introduce confounding by disease severity. This limitation is inherent to observational database studies, as symptom severity is not directly captured in administrative data. To mitigate this, we applied propensity score matching (PSM) to balance comorbidities and other proxies of disease burden between groups, achieving excellent covariate balance (all standardized mean differences <0.1). Furthermore, the use of an active comparator design helps reduce bias related to treatment indication.

Moreover, negative control outcomes were incorporated to detect potential residual bias and assess whether the observed associations could be explained by systematic error rather than true effects. Nonetheless, we acknowledge that residual confounding due to unmeasured clinical severity remains possible and have noted this in the Discussion.

3. Negative control outcomes: A more detailed explanation of the negative control outcomes would be helpful. Were these conditions compared between patients using antidepressants and those not using them? Additionally, the manuscript refers to supplemental Table 4d—could you clarify its location or provide it if missing?

Response: We appreciate the reviewer's comment and the opportunity to clarify. Negative control outcomes were included to assess potential residual bias or spurious associations in our analysis. These outcomes represent conditions that are biologically and clinically unrelated to antidepressant use and therefore should not differ between treatment groups if confounding is adequately controlled.

In our study, we compared the incidence of several negative control conditions such as blepharitis, frostbite, acute appendicitis, nummular dermatitis, eczematous dermatitis of the eyelid, retinal detachment, and protozoal diseases between IBS patients using antidepressants and those not using them. As expected, no significant differences were observed, supporting the robustness and internal validity of our findings.

The results of these analyses are presented in Supplementary Table 1d, which includes the negative control outcome comparisons. We will ensure that this table is clearly labeled and properly referenced in the final version of the manuscript.